# Heliox at 4 ATA Reduces Error Rates Compared to Trimix and Air, but It Does Not Affect Short-Term Memory in Hyperbaric Conditions

**DOI:** 10.3390/biology14121748

**Published:** 2025-12-05

**Authors:** Rita I. Sharma, Natalia D. Mankowska, Anna B. Marcinkowska, Pawel J. Winklewski, Jacek Kot

**Affiliations:** 1Department of Neurophysiology, Neuropsychology and Neuroinformatics, Medical University of Gdansk, 80-210 Gdansk, Poland; pawel.winklewski@gumed.edu.pl; 2Applied Cognitive Neuroscience Lab, Department of Neurophysiology, Neuropsychology and Neuroinformatics, Medical University of Gdansk, 80-210 Gdansk, Poland; natalia.mankowska@gumed.edu.pl (N.D.M.); anna.marcinkowska@gumed.edu.pl (A.B.M.); 32nd Department of Radiology, Medical University of Gdansk, 80-210 Gdansk, Poland; 4National Centre for Hyperbaric Medicine, Institute of Maritime and Tropical Medicine in Gdynia, Medical University of Gdansk, 80-210 Gdansk, Poland; jkot@gumed.edu.pl

**Keywords:** hyperbaric cognition, heliox, trimix, nitrogen narcosis, diving safety

## Abstract

Cognitive impairment during diving is a well-known disturbance that can be reduced by the use of various breathing mixtures depending on the type of diving. However, hyperbaric conditions also occur outside the underwater environment in hyperbaric chambers, which are commonly used for therapeutic purposes. Besides medical attendants in hyperbaric chambers, divers in underwater conditions are primarily exposed to hyperbaric effects on cognition. Based on the available literature, inert gas narcosis symptoms occur in most cases at depths equivalent to 4 atmospheres absolute (ATA). We examined healthy volunteers using neuropsychological tests to assess reaction time, memory, and number of errors during task completion. Our aim was to analyse the severity of cognitive disturbance in hyperbaric conditions at a pressure of 4 ATA depending on the breathing mixture (air, trimix, heliox) at three different stages: in normobaric conditions, at 4 ATA and after hyperbaric conditions ceased. The results showed that using heliox as a breathing mixture reduces error rate in hyperbaric conditions. This finding leads to the conclusion that the use of heliox may be associated with greater safety for divers, even during recreational diving (up to the equivalent of 5 ATA), as well as for medical attendants in the chamber.

## 1. Introduction

The increasing popularity of recreational diving has led to a growing number of individuals being exposed to hyperbaric conditions during leisure activities. As participation rises, greater emphasis must be placed on divers’ safety and the prevention of accidents. According to the DAN Europe Database [1], risk factors and individual predispositions for diving-related cognitive impairment are not sufficiently understood. Among the physiological hazards, nitrogen narcosis is one of the most extensively described phenomena influencing cognitive performance. It has been shown to occur already at depths over 10 metres of seawater (msw; equivalent to 2 ATA) [2,3], manifesting as disturbances of attention, euphoria, slowed reaction time, and difficulties in rapid decision-making [4,5]. Nearly 90 years ago, researchers noted that narcosis develops while breathing compressed air, but it is much less evident when nitrogen is replaced by helium [4]. When participating in recreational diving (deeper than 40 msw), divers are recommended to breathe mixtures such as trimix (21.6% air, 36.7% nitrogen, and 41.7% helium) and heliox (21% air and 79% helium) to reduce nitrogen narcosis. However, how inert gas narcosis affects cognitive functioning has not been fully explained [4].

There is scarce empirical evidence concerning cognitive functioning during recreational dives. Zhao et al. [6] reported that a single exposure to 2 ATA impaired processing speed and memory in young divers, while animal models have demonstrated that prolonged exposures could trigger neuroinflammation and subsequent cognitive decline. Brebeck et al. [7] tested 108 experienced divers at 24 msw (3.4 ATA) in open water conditions. Participants breathing nitrox showed superior long-term memory outcomes compared with those breathing air, while short-term attention and number-connection performance remained unaffected [8].

Fothergill et al. [9] examined the combined effects of increased partial pressures of nitrogen and carbon dioxide (pN_2_ and pCO_2_, respectively). They demonstrated that hypercapnia delayed task performance but preserved accuracy, whereas nitrogen narcosis impaired both performance speed and accuracy. Similarly, Williamson et al. [10] showed that in occupational shallow-water divers, working memory errors increased with age and exposure time, while reaction speed and sustained attention were influenced by maximum diving depth. Philp et al. [11] further confirmed significant impairments in immediate recall during chamber dives to 36 msw (4.6 ATA), despite no long-term differences after decompression. Collectively, these findings confirm that acute cognitive disturbances do occur during diving and may depend on the exposure profile, depth, breathing gas mixture, and diver-specific factors such as age and gender.

Among well-known factors mentioned above, we cannot ignore the influence of inert gases, particularly nitrogen, which exerts a dose-dependent narcotic effect that worsens as the partial pressure increases. Beyond narcosis, oxygen also plays a dual role: short-term hyperoxia has been associated with transient cognitive enhancement in normobaric conditions, while a high oxygen pressure in hyperbaric conditions may provoke neurotoxic effects with a narrow therapeutic window [12,13]. Similarly, CO_2_ retention, promoted by increased gas density and respiratory workload, strongly modulates cerebral blood flow and can exacerbate narcotic effects [13]. Thus, differences in breathing gas mixtures—for example, the lower density of heliox compared with air—may indirectly improve cognitive outcomes by reducing CO_2_ retention.

Simple, portable, and standardised methods are required to measure these cognitive disturbances. Neuropsychological assessment provides such tools, including widely validated tests for working memory, attention, inhibition, and processing speed (e.g., the Digit Span and Corsi Block Tapping tasks, the Stroop test, the Trail Making Test, and the Sternberg Memory Test) [14]. These tests are suitable both for laboratory-based hyperbaric chamber studies and open water research, making them valuable for investigating the transient cognitive changes associated with diving. However, it has also been well established that cognition is influenced by stress, fatigue, emotional state, gender, and age [15]. Some of these confounding factors can be minimised by narrowing the inclusion criteria (e.g., age 18–55 years) [16].

Beyond diving, research into oxygen-related environments underscores the complexity of brain responses to altered gas mixtures. Hyperbaric oxygen therapy has shown therapeutic promise in reducing neuroinflammation and improving cognition in conditions such as dementia and traumatic brain injury [17,18,19,20]. Conversely, hypoxic environments, such as high altitude, have been linked to deficits in attention and executive functioning, highlighting the detrimental effects of insufficient oxygen delivery [21,22,23,24,25]. These contrasting findings illustrate that oxygen availability and gas composition are pivotal determinants of cognitive performance across diverse environmental settings. Taken together, the literature suggests that hyperbaric exposure, at depths relevant to recreational diving, impairs cognitive functioning, but the severity and profile of impairment vary with the breathing gas mixture. Despite decades of research, there have been very limited systematic intra-individual comparisons of cognitive performance across compressed air, trimix, and heliox exposures in controlled hyperbaric chamber settings. Moreover, the variability and stability of results across normobaric and hyperbaric conditions are seldom analysed, yet this information is crucial for distinguishing true environmental effects from measurement variability.

To address this knowledge gap, we aimed to conduct a comparative analysis of standardised neuropsychological test results obtained from the same individuals under normobaric and hyperbaric conditions and to evaluate the effects of three breathing gas mixtures—air, trimix, and heliox—on cognitive functioning during simulated dry dives at 4 ATA (equivalent to 30 msw). We hypothesised that the severity of cognitive impairment in hyperbaric conditions (at 4 ATA) depends on the breathing gas mixture. Specifically, we hypothesised that the greatest reduction in performance occurs while breathing compressed air, an intermediate reduction occurs when breathing trimix, and the least pronounced reduction occurs while breathing heliox. Our study should provide empirical evidence on how hyperbaric exposure and the breathing gas mixture interact to shape cognitive functioning. We expect our findings to enhance theoretical models of inert gas narcosis and contribute to practical recommendations for diver safety and hyperbaric medicine.

## 2. Materials and Methods

### 2.1. Participants

The study was conducted on young, healthy volunteers of both sexes. The inclusion criteria were age 18–55 years, good general health, no overuse of alcohol or psychoactive drugs, and the ability to provide written informed consent following a qualifying interview. The exclusion criteria were age >55 years, uncontrolled cardiovascular disease, a history of thoracic or otolaryngologic surgery, obstructive pulmonary disease, and pregnancy.

The information about our study was widely disseminated at our university via mass media, among divers cooperating with the National Centre for Hyperbaric Medicine, and among participants of Hyperbaric Medical Training organised at the aforementioned centre. The volunteers were familiarised with exclusion criteria and spoke to the researchers about the course of the study.

A total of 71 volunteers (32 men and 39 women) were recruited. Of these, 46 met the inclusion criteria and were included in the study. The 25 volunteers who were not included either did not meet the requirements or declined to accept the study conditions. During the study, seven participants withdrew without providing a reason. Data from six participants were excluded from the analyses because they completed only a single test, which could not be evaluated in the context of the study hypotheses. The mean (*M*) age of all enrolled volunteers was 32.5 years (*SD* = 8.38, *Me* = 34, range: 19–46 years old).

The aforementioned participant’s withdrawal from the study was mainly due to scheduling conflicts and study demands. As these factors were unrelated to performance, they are unlikely to have biased the findings, though the smaller sample may have slightly reduced statistical power.

### 2.2. Procedure

Each volunteer was asked to attend three separate sessions at the National Centre for Hyperbaric Medicine in Gdynia (Poland), where the study was conducted. During each session, dry hyperbaric conditions were achieved in a multiplace hyperbaric chamber at a pressure of 4 ATA (equivalent to 30 msw), and the participants performed the psychometric test battery at three time points:Stage I (before)—prior to entering the chamber (1 ATA, ambient air);Stage II (4 ATA)—during exposure at 4 ATA while breathing one of the experimental gas mixtures;Stage III (after)—following decompression back to 1 ATA (ambient air);The breathing gas mixtures administered at 4 ATA were:Air (20.8% oxygen, 78% nitrogen);Trimix (21.6% oxygen, 36.7% nitrogen, 41.7% helium);Heliox (21% oxygen, 79% helium).

Each participant was tested with all three mixtures in a randomised order, with a mandatory 24 h interval between each exposure. The participants were blinded to the mixture during each trial; only the chamber operator and attendant knew the composition. Information about the gas used was provided to participants upon request after completing all stages.

Because of the increased pressure, the chamber temperature during exposures ranged from 31 to 37 °C (*M* = 34.53, standard deviation [*SD*] = 1.49), with no significant differences between the gas conditions. In total, 40 participants completed the experiment with air, 36 with heliox (17 women), and 37 with trimix (18 women). Consequently, not all results were available for statistical analyses.

During each compression, a maximum of two participants was present in the chamber, accompanied by an attendant. The participants were seated 1.5 m apart, with test displays positioned 1 m in front of them. The mean compression rate was 5–6 msw/min, adjusted individually according to the participants’ ability to equalise middle-ear pressure. Communication during testing was maintained through headsets with integrated microphones. Each participant was provided with a wireless mouse and mouse pad, as well as pen and paper. A wireless router positioned outside the chamber ensured stable internet connectivity for the online tasks. The exposure time at 4 ATA was 25–30 min. The decompression was carried out in accordance with decompression tables of the Polish Ministry of Health (Regulation of 17 September 2007, on health conditions for underwater work) for the worst-case scenario, assuming the longest decompression for a specific time. No emergency decompression procedures were required across 59 completed compressions.

#### Neuropsychological Assessment

To comply with safety standards in the hyperbaric chamber, the cognitive tasks were administered on oil-immersed underwater tablets (Alltab 4.0, Valtamer, Helsinki, Finland), adapted for safe use under hyperbaric conditions. The responses were registered using wireless computer mice. The tasks were programmed in the PsyToolkit software (version 3.4.6) [26,27] and delivered online.

The test battery was designed to be concise (with a completion time of 10–15 min) while covering key aspects of cognitive functioning. It included standardised tasks assessing working memory, attention, inhibitory control, and processing speed. Although not exhaustive, this set allowed for repeated assessment across multiple exposures.

Prior to chamber entry, each participant received standardised instructions and a practice trial for every task to ensure comprehension. The total testing time per session was approximately 15 min, depending on individual pace. Each session began with instructions, followed by a short trial phase, and then the formal testing sequence (see Figure 1).

### 2.3. Ethics

This study was conducted in accordance with the principles of the Declaration of Helsinki (2013 revision) and was approved by the Ethical Committee of the Medical University of Gdansk (approval no. 242/2020). All participants provided written informed consent prior to inclusion and were informed of their right to withdraw from the study at any time without providing a reason.

### 2.4. Cognitive Assessment

Cognitive performance was evaluated with a battery comprising three psychometric tasks: the Digit Span, Corsi Block Tapping, and Simon tasks (Table 1).

#### 2.4.1. The Digit Span and Corsi Block Tapping Tasks

Both tasks require the participants to memorise sequences and to reproduce them either in the same order (forward trials) or in the reverse order (backward trials). Forward trials primarily assess immediate memory and attention, while backward trials additionally engage working memory capacity.

**Digit Span task:** Digits were presented sequentially at a rate of one per second. After the sequence ended, the participants reproduced the sequence using an on-screen keypad containing the digits. This task measures verbal memory.

**Corsi Block Tapping task:** The test used a digital board displaying white squares arranged according to the original layout described by Kessels [28]. During sequence presentation, individual squares turned green one at a time. After the sequence ended, the participants reproduced the order by clicking the corresponding squares with a mouse. This task measures visuospatial memory [28,29,30].

In both tasks, the sequence length increased progressively from two to nine elements, depending on the participant’s performance. If two consecutive sequences of the same length were recalled incorrectly, the task was terminated. The maximum sequence length correctly recalled was recorded as the participant’s score.

#### 2.4.2. The Simon Task

The Simon task was administered to assess inhibitory control and the ability to resolve cognitive interference arising from conflicting spatial cues [24]. The stimuli consisted of coloured squares presented on either the left or right side of the screen. The participants were instructed to respond only to the colour of the stimulus (left mouse click for green; right mouse click for red), ignoring its spatial location. The performance outcomes were:Reaction time (ms);Accuracy (% correct responses);The Simon effect, calculated as the mean RT (reaction time) difference between incongruent and congruent trials.

### 2.5. Data Preparation and Statistical Analysis

Prior to statistical analysis, the data were sorted and filtered. Given the exploratory nature of the study and the predefined hypotheses, individual outliers were not removed from the dataset.

For the Simon task, the initial dataset comprised 16,350 responses (50 per participant in each trial). Incorrect responses (*n* = 306) and responses with latencies greater than 5 s (*n* = 13) were excluded, as it was not possible to determine their validity. Consequently, 16,031 responses (98.05%) were included to calculate the RTs. The frequency and distribution of incorrect responses were analysed separately. For each participant, the median RTs were calculated for the congruent and incongruent trials in each of the three breathing conditions (air, heliox, and trimix) across the three experimental stages (before, 4 ATA, and after). The median was used instead of the mean because the RT data had a non-normal distribution. The Simon effect was computed for each condition as the difference between the median RT in incongruent and congruent trials.

For the Corsi Block Tapping and Digit Span tasks, five values equal to zero were identified in the dataset. These values were treated as missing data, as it was unclear whether they reflected technical errors or task-related issues.

Statistical analyses were carried out using IBM SPSS Statistics 29.0.0.0 (IBM Corp., Armonk, NY, USA). All numerical data were tested for normality using the Shapiro–Wilk test. Because the distributions of the RTs and the sequence reproduction scores deviated from normality, nonparametric tests were applied for statistical analysis. The repeated-measures Friedman test was applied for comparisons across multiple experimental conditions. Where appropriate, the Wilcoxon signed-rank test was used for post hoc pairwise comparisons. Statistical significance was set at *p* < 0.05. To maximise data retention, analyses were performed with available cases per condition (pairwise deletion), which led to slight variations in the sample size across comparisons.

## 3. Results

First, we compared the RTs obtained in the congruent and incongruent trials in the Simon task. There were significant differences in all conditions (all *p*s < 0.01), meaning that the subjects needed more time to respond to conflict trials.

Table 1 presents the descriptive statistics for the cognitive tests. This table includes detailed information on the sample size (*N*) available for each experimental condition and measurement stage. The reported *N* reflects all data available for a given test. However, it should be noted that the number of observations included in specific statistical analyses may vary slightly due to the exclusion of cases with missing data in pairwise comparisons. We have noted these differences in *N* in the text.

### 3.1. General Influence of Hyperbaric Exposure on Cognitive Functioning

To assess whether hyperbaric conditions have a reversible impact on cognitive functioning, we compared performance across the three stages: before exposure (before), during exposure (4 ATA), and after decompression (after). We analysed the results obtained in the three experiments using different breathing gas mixtures separately with the repeated-measures Friedman test, followed by the Wilcoxon signed-rank test for post hoc comparisons.

#### 3.1.1. Reaction Time (RT)

The Friedman test showed significant differences in RTs among the study stages (all *p*s < 0.001). Thus, we conducted post hoc tests to identify the sources of these differences. Under all conditions, the RTs were the shortest during the after stage. In each of the three groups, the RTs differed significantly between the before and after stages and between the after and 4 ATA stages. The differences in the RTs between the before and 4 ATA stages were less notable and applied only to the incongruent trials. When breathing heliox during the 4 ATA stage, the RTs for the incongruent trials were much longer than the initial ones (*p* = 0.022). The result for the incongruent trials when breathing trimix was close to statistical significance (*p* = 0.052), with *M*_before_ of 709.02 ms and *M*_4 ATA_ of 754.50 ms (for *N* = 32). There were no significant differences in the RTs between the before and 4 ATA stages when breathing air. Although in the experiments with air and trimix, there was an increase in the RTs when performing the test at 4 ATA compared with the initial values, these differences were not significant. We also found no significant differences in the Simon effect among the three stages. Figure 2 illustrates the differences in RTs and error rates between the congruent and incongruent trials across the breathing gas mixtures and stages.

For the congruent trials, the error rates across different conditions were relatively consistent in most cases, ranging from 1% to 1.65%. We observed the highest error rates during the before stage and while breathing air, with 2.10% and 2.48%, respectively. In contrast, the incongruent trials showed somewhat greater variability in the error rates. When breathing air, the error was 2.3% during the before stage and then nearly doubled to 4.38% during the 4 ATA stage, followed by a decrease to 2.77% during the after stage. For heliox, these differences were smaller: the error rate was 1.89% for the before stage, it decreased slightly to 1.12% during the 4 ATA stage, and then increased to 2.46% during the after stage. While breathing trimix, the error rate was 1.41% during the 4 ATA stage, compared with 1.36% during the before stage and 1.55% during the after stage. 

#### 3.1.2. Attention and Memory Performance

Based on the Friedman test, the difference in the length of the sequences reproduced in the Corsi Block Tapping forward task when breathing air was close to statistical significance (*p* = 0.057). According to an additional pairwise comparison, the mean length of the sequence reproduced under hyperbaric conditions (*M* = 5.77) was significantly lower than the mean after decompression (*M* = 6.31) (*N* = 39; *z* = –2.131, *p* = 0.033). There were no significant differences when the participants breathed heliox or trimix. Additionally, there were no differences in the Corsi Block Tapping backward task or the Digit Span task. Figure 3 illustrates the average length of reproduced sequences across all tasks, categorised by the experimental stages and breathing gas mixtures.

### 3.2. Differential Effects of Breathing Gas Mixtures

Next, we compared performance between the three breathing gas mixtures (air, trimix, and heliox) at each time point.

#### 3.2.1. Reaction Time (RT)

We conducted a separate Friedman test for each trial type to explore whether the type of breathing gas mixture had a differential effect on performance in the congruent and incongruent trials. The initial RTs (*N* = 34) obtained in each of the experiments differed significantly from each other during the incongruent trials (*χ*^2^(2) = 9.235, *p* = 0.010). For the congruent trials, there was a trend towards statistical significance (*χ*^2^(2) = 5.353, *p* = 0.069). Post hoc analyses indicated that these differences were mainly related to the RTs obtained when breathing air, which were the highest. These differences were particularly noticeable for the incongruent trials (air vs. heliox: *N* = 36, *z* = –2.168, *p* = 0.030; air vs. trimix: *N* = 35, *z* = −2.784, *p* = 0.005). For the congruent trials, the RTs for the before stage when breathing air were significantly higher than those when breathing trimix (*N* = 35, *z* = –2.539, *p* = 0.011), but there was no difference between air and heliox or between heliox and trimix. There were no differences in RTs between the breathing gas mixtures for the 4 ATA and after stages. In addition, the size of the Simon effect did not differ under any of the conditions.

Taken together, these findings indicate that the differences in the RTs during the before stage across the breathing gas mixtures—in particular, the higher RTs when breathing air—may confound the interpretation of the changes observed during the subsequent stages. Therefore, to control for these baseline differences and to better isolate the effects of hyperbaric exposure, we converted the participants’ RTs to percentages to standardise the data and allow for comparison of relative changes in subsequent stages of the study. We considered the before-stage values as a reference (100%) and then determined the values for the 4 ATA and after stages. We transformed the data separately for each participant and condition (task and type of the trial, breathing gas mixture, and study stage). This resulted in changes in RTs as relative deviations from the individual baseline, which allowed for a better visualisation of the impact of hyperbaric conditions regardless of baseline differences between participants and conditions. Thus, we present RTs in Figure 4 in two formats: absolute values in milliseconds (A) and normalised percentages relative to baseline (B). This dual presentation facilitates comparison of the raw and standardised results across the various breathing gas mixtures.

When breathing air, the changes between the before and 4 ATA stages were the smallest and nonsignificant compared with the other breathing gas mixtures. When breathing heliox, the RTs for the incongruent trials increased significantly from baseline (*N* = 32; *z* = −2.412, *p* = 0.016), and when breathing trimix, these differences were close to statistical significance (*N* = 32; *z* = −1.945, *p* = 0.052). Notably, the RTs for these two conditions were very similar (see Figure 4B). The RTs for the congruent trials did not change significantly between the before and 4 ATA stages. The greatest percentage changes in the RTs occurred when comparing the after stage with the two previous stages. These changes were consistent across all breathing gas mixtures, with statistical analyses confirming no significant differences between the mixtures.

#### 3.2.2. Attention and Memory Performance

The Friedman test indicated significant differences only for the Digit Span forward task executed during the after stage (*n* = 35, *χ*^2^(2) = 6.813, *p* = 0.033). Based on the Wilcoxon test, these differences were due to significantly higher results for the test when breathing trimix compared with breathing air (*M*_trimix_ = 6.67 and *M*_air_ = 6.25; *N* = 36, *Z* = −2.073, *p* = 0.038) (see Figure 3).

## 4. Discussion

The most important and novel finding of the present study is that when compared with breathing air and trimix, breathing heliox at 4 ATA was associated with the lowest number of errors in executive interference trials, even with a modest increase in the RTs. This effect was most clearly observed in the Simon task, where incongruent stimuli require inhibition of automatic responses and engagement of higher-order control processes. The incongruent error rate nearly doubled when breathing compressed air at 4 ATA, breathing trimix resulted in only negligible changes, and breathing heliox markedly reduced the error rate. After decompression, the error rate increased again, returning to what was observed during the before stage. This distinctive pattern underscores that heliox promotes accuracy under cognitive conflict, even if accompanied by slower responses. From an operational perspective—particularly in diving, where procedural mistakes carry disproportionate risks—this accuracy-preserving profile is of greater value than preserving speed at the expense of errors. Thus, the heliox-related reduction in mistakes represents the most practically relevant cognitive effect observed in our study.

Diving safety depends more on avoiding critical errors than on maintaining maximal speed. Underwater mistakes such as misinterpreting signals, skipping checklist steps, or operating the wrong valve may immediately compromise safety, while minor delays in reaction time are generally tolerable. Our results suggest that heliox supports a strategic shift in cognitive control: participants accept slower responses, but accuracy is preserved. This is consistent with the speed–accuracy trade-off framework, in which decision thresholds are adjusted more conservatively [31]. In contrast, breathing air favours a more liberal, speed-preserving policy, at the cost of an increased probability of making an error. Trimix, which combines helium and nitrogen, showed an intermediate pattern, with minimal changes in errors and a moderate increase in the RTs. Similar conclusions come from other research [32,33,34]; however, it should be emphasised that most studies using nitrogen and helium as breathing mixtures focus on diving to deeper depths than recreational diving, while our study was conducted at 4 ATA (equivalent to 30 msw).

This accuracy-preserving profile aligns with both classic and modern accounts of nitrogen narcosis, in which accuracy and judgement degrade as pN_2_ rises [3]. Given that helium is far less narcotic than nitrogen, this burden is reduced, and the lower density of helium decreases ventilatory workload, mitigating CO_2_ retention. Carbon dioxide retention, which disturbs cognitive performance [7,35,36,37], has to be taken into account in “dry diving” assuming the hyperbaric chamber functions as a semi-closed circuit with variable ventilation [38].

Together, these properties may explain why heliox supports a cautious, accuracy-focused response style under hyperbaric conditions [4,5]. The heliox accuracy advantage can be explained by several interacting mechanisms. First, and central, is the reduced narcotic load. Nitrogen narcosis impairs performance speed and accuracy, particularly in tasks demanding inhibitory control. Substituting helium for nitrogen lowers pN_2_, thereby reducing cortical noise and improving information integration [39]. Our finding that the error rate decreased when breathing heliox at 4 ATA is consistent with this mechanism. Second, gas density and ventilatory effort matter. Heliox is significantly less dense than air, which lowers the work of breathing and reduces CO_2_ accumulation. Hypercapnia delays responses and exacerbates narcosis [9]; thus, heliox may protect accuracy by maintaining a more favourable respiratory and metabolic state. Third, while all three breathing gas mixtures contain ~21% oxygen, and thus the oxygen partial pressure is similar at 4 ATA, the interaction between oxygen and helium versus the interaction between oxygen and nitrogen might subtly affect cortical excitability and arousal [40]. In addition, hyperoxia, which may occur in hyperbaric conditions depending on the breathing gas used, prolongs the reaction time [41]. We cannot exclude such modulatory effects.

Across all conditions, the RTs were shortest during the after stage (i.e., following decompression), reflecting strong practice and state effects. Interestingly, the Simon effect is subject to a learning curve, with the effect reducing over time, but it returns [42].

The participants grew more familiar with the demands of the tasks because they had performed the same test each time, and the relief they felt following the most demanding stage (compression and exposure) likely reduced stress and arousal. This explains why the post-decompression results showed not only recovery but even improvement beyond baseline. This finding is consistent with previous studies demonstrating that impairments observed during hyperbaric exposure are largely reversible upon ascent [10,11,41]. The fact that accuracy was maintained while breathing heliox during the 4 ATA stage further emphasises the robustness of its effect.

Unlike interference control, short-term memory span (i.e., measured with the Digit Span and Corsi Block Tapping tasks) remained largely stable across stages and mixtures, with only isolated differences. For example, the Corsi Block Tapping task forward scores while breathing air were slightly lower during the 4 ATA stage than during the after stage, while the Digit Span task forward scores were higher during the after stage when breathing trimix compared with air. However, such sporadic findings are difficult to interpret and may reflect practice or sample variability rather than systematic hyperbaric effects. These results suggest that short-term memory and basic attention are not strongly impaired by short-term exposure at 30 msw. This finding differs from previous studies that have reported memory decrements under similar pressures [11,43], but those studies often used more demanding paradigms (e.g., free recall and the Sternberg task), which are more sensitive to lapses. Our battery, designed for chamber safety and brevity, may have lacked sufficient sensitivity to detect subtle changes.

Our findings are consistent with several patterns established in the literature. First, they reinforce that cognitive functioning at recreational-range depths is not globally impaired, but specific functions—particularly executive interference control—can show modest, reversible changes [7,9,11]. Second, they highlight that the breathing gas mixture composition influences performance: air is associated with higher error rates, whereas helium-based mixtures promote accuracy. This corresponds with helium’s lower narcotic potency and gas density [34,44]. Our results also align with prior work demonstrating that heliox can enhance accuracy in interference tasks. Lee et al. [34] showed that divers breathing heliox at 3.6 ATA performed Stroop tasks with improved accuracy compared with breathing compressed air. These authors concluded, as we do, that helium substitution may directly mitigate the effects of nitrogen narcosis [45].

We did not find robust differences in mean RTs or memory scores between the breathing gas mixtures, which contrasts with studies reporting superior memory outcomes when breathing trimix or heliox [7,34]. Such discrepancies may be due to methodological differences in the exposure duration, task sensitivity, and participant expertise. Our study suggests that the breathing gas mixture effects are most evident in error control under conflict, rather than in general processing speed or simple span performance. Historical heterogeneity further supports this view. For example, Bennett et al. [46] reported an 18% increase in errors at 4 ATA, whereas Frankenhaeuser et al. [22] found no differences, and Kiessling and Maag [47] observed ~20% slowing at similar pressures. Steinberg and Doppelmayr [39] noted selective impairments of inhibitory control at 3 ATA. Collectively, these studies suggest that the outcomes depend heavily on task type, depth, exposure time, and diver characteristics. Our findings provide nuance by showing that breathing gas mixture-specific effects may primarily shape strategic allocation of speed versus accuracy.

Additionally, among the volunteers, some were recreational divers, while others were first exposed to hyperbaric conditions. To determine whether diving experience influences cognition, Kowalski et al. [48] examined twenty-three professional military divers and twenty-three healthy subjects without diving experience. Results of reaction time and error rate in both groups showed that reaction times were higher in the group of experienced divers, but error rate increase was not observed in either group.

Undoubtedly, underwater conditions vary from conditions in a hyperbaric chamber. Immersion, ambient temperature, equipment, and experience differentiate individuals’ anxiety and motivation. Our study was conducted in a hyperbaric chamber (dry diving); therefore, factors like immersion, water temperature, and buoyancy did not affect our results. Baddeley [49] described that generalising results obtained in hyperbaric chambers to the open water environment (wet diving) is not entirely justified. Hobbs [43] performed a digit-letter substitution test among 125 divers in underwater conditions (37–45 msw) and on the surface. Divers reporting anxiety underperformed on the task underwater compared to those not reporting anxiety. The effect of pressure on task performance in dry conditions was less compared to open water, meaning that in the hyperbaric chamber, the time needed to complete the task was shorter than in underwater conditions.

### Limitations

There are several limitations to this study that temper the interpretation of our findings. First, we did not record physiological parameters (e.g., end-tidal CO_2_, peripheral oxygen saturation [SpO_2_], or respiratory effort). Without these, we cannot disentangle whether the accuracy advantage while breathing heliox stems primarily from reduced narcosis, lower CO_2_ retention, or both. In addition, all breathing gas mixtures caused hyperoxia at 4 ATA. We did not consider the synergic effects of hyperoxia on cognitive functioning [22,50].

Second, the cognitive battery was necessarily brief due to chamber constraints. Additionally, the exposure time was limited, so we could not extrapolate the results to longer exposure times. Performing several tasks in a row could assess the impact of time, functional deteriorations, or cognitive adaptation. While practical, this approach would have limited sensitivity to subtle decrements. Additional tasks assessing psychomotor vigilance, visuomotor coordination, and verbal recall would provide greater ecological validity. What is more, among the factors influencing reaction time, we enlist age, gender, fatigue, hydration, distraction, limb used for the test, and type of stimulus [51,52,53]. Our subjects were asked to arrive for the tests in a relaxed state and were instructed to avoid stimulants, including alcohol, for 24 h prior to the test. Also, to eliminate the risk factors of decompression sickness, like dehydration [54], each subject received still water for optimal hydration before starting the test. During exposure while performing the second stage of the test in the chamber, some volunteers reported a violent need to urinate. As a stress factor, we cannot exclude such incidents as factors leading to deliberately completing the tasks faster than usual. Also, despite the fact that participants wore audio headsets to stay in contact with the researcher, the noise inside the chamber could have been a distraction for participants sensitive to auditory stimuli [55].

Third, we observed some practice and state effects, reflecting the natural learning curves inherent in repeated-measures designs [56]. The participants received instructions regarding the test before the first test and performed the test a total of three times on the same day. It should be emphasised that tests using a particular breathing mixture were performed several days apart, so we consider the risk of a learning curve to be minimal, but not entirely impossible. Such influences are virtually unavoidable in this type of experimental paradigm and, while present, are unlikely to account for the mixture-specific differences reported here.

Finally, our study was conducted in a dry hyperbaric chamber. Although such environments permit controlled experiments, the results cannot be generalised directly to open water diving, where immersion, buoyancy, temperature, and equipment impose additional stressors [49]. Previous research has shown that manual dexterity deteriorates to a greater extent in wet dives than in dry chambers, partly due to added physical constraints [49,50].

## 5. Conclusions

Despite inevitable methodological limitations, the practical implications of our study are clear. At a depth of 30 msw and for short-term exposure, overall cognitive functioning remains largely preserved. However, the quality of performance differs depending on the breathing gas mixture. When breathing compressed air, divers tended to maintain speed at the expense of an increase in the number of errors. In contrast, breathing heliox promoted a more conservative response strategy, with slightly longer RTs but markedly fewer mistakes. Given that in diving or medical attendant safety contexts errors typically pose greater risks than minor delays, this accuracy-preserving effect makes heliox the preferable breathing gas mixture from an operational safety perspective. This advantage complements heliox’s well-documented physiological benefits in reducing both nitrogen narcosis and ventilatory burden. From a practical standpoint, dive planning and training should place stronger emphasis on accuracy rather than speed, with checklists and operational procedures designed to minimise errors. Short-term memory deteriorated at 4 ATA in all trials; however, the type of gas mixture did not significantly affect statistical relevance. Therefore, the results do not allow us to conclude that short-term memory, as a cognitive function, is less impaired when using heliox. Pre-dive rehearsal and cognitive warm-ups may further help to stabilise performance and to reduce variability linked to practice effects. In conclusion, short hyperbaric exposure (up to 30 min) at 4 ATA induces only modest and reversible cognitive changes, with executive timing under conflict appearing to be the most sensitive domain. The central finding of this study is that heliox minimises cognitive errors at depth, whereas air increases them. This suggests that heliox supports a cautious, accuracy-first decision strategy, while air promotes speed at the expense of precision. For diving safety—where mistakes are more critical than transient slowing—this accuracy advantage of heliox represents a meaningful operational benefit, strengthening its role in both mitigating narcosis and enhancing cognitive reliability under pressure.

## Figures and Tables

**Figure 1 biology-14-01748-f001:**
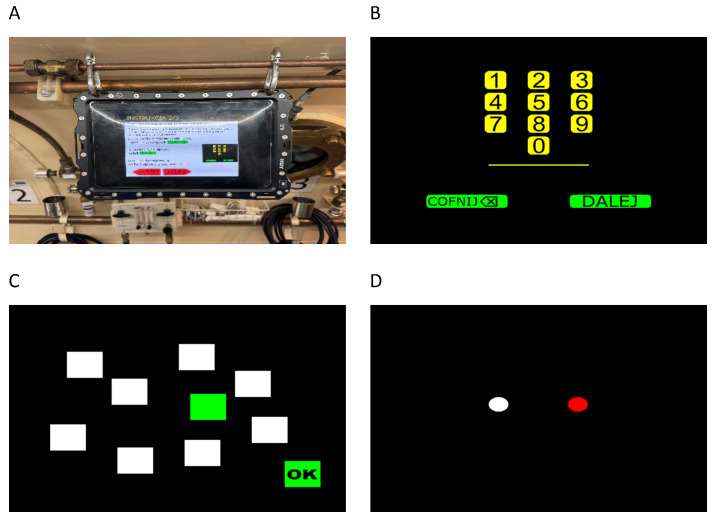
Test instructions and tasks used in neuropsychological assessment: (**A**) oil-immersed tablet used in the test, (**B**) the Digit Span task, (**C**) the Corsi Block Tapping task, and (**D**) the Simon task.

**Figure 2 biology-14-01748-f002:**
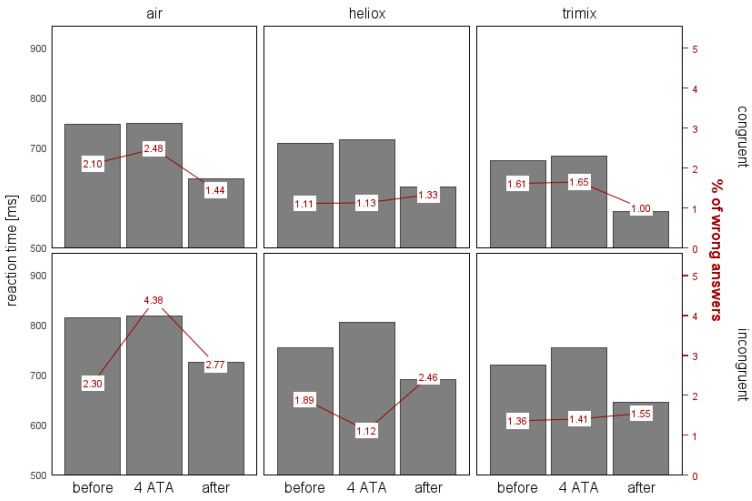
The mean reaction times (ms) and the error rates in the congruent and incongruent trials of the Simon task, presented by the experimental stage and gas mixture.

**Figure 3 biology-14-01748-f003:**
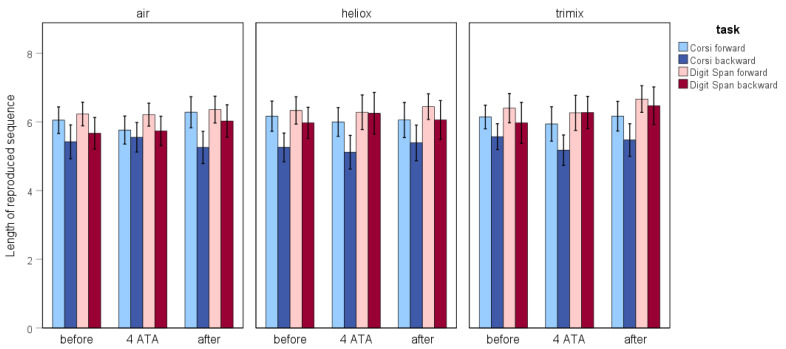
The average length of the reproduced sequences across the Corsi Block Tapping and Digit Span tasks, shown by testing stage and gas mixture.

**Figure 4 biology-14-01748-f004:**
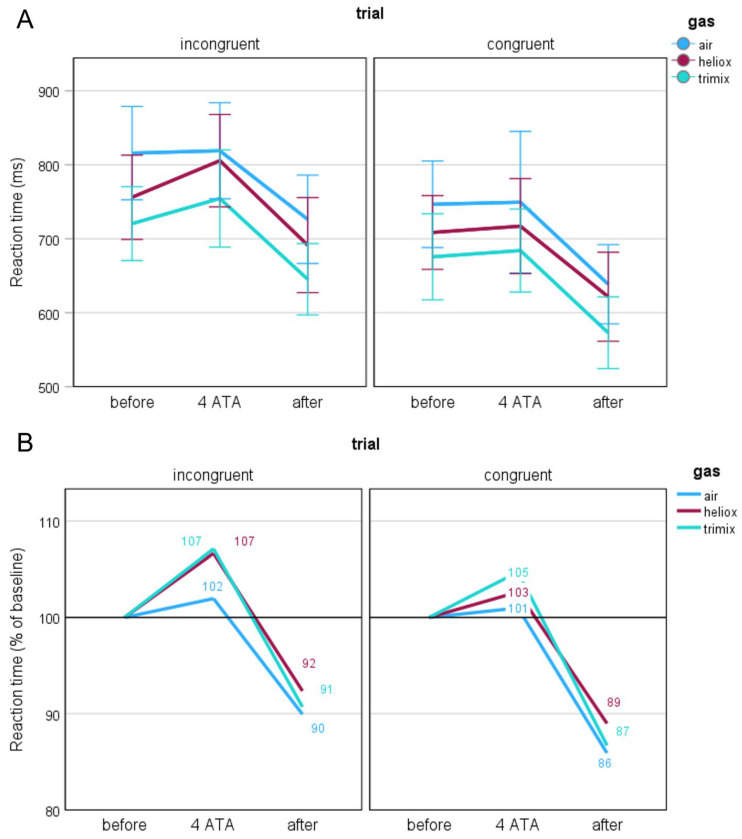
Reaction times presented as raw values in milliseconds (**A**) and as percentages relative to baseline (**B**) across different gas mixtures and experimental stages.

**Table 1 biology-14-01748-t001:** Results of cognitive tasks at all stages of the experiment with different gas mixtures.

Task			Air			Heliox			Trimix		
			Before	4 ATA	After	Before	4 ATA	After	Before	4 ATA	After
Corsi Block Tapping task	Backward	*N*	38	38	39	35	34	36	35	34	36
		*M*	5.421	5.553	5.256	5.257	5.118	5.389	5.571	5.176	5.472
		*SD*	1.500	1.309	1.446	1.221	1.409	1.536	1.092	1.267	1.404
	Forward	*N*	39	38	39	36	32	36	35	34	36
		*M*	6.051	5.763	6.282	6.167	6.000	6.056	6.143	5.941	6.167
		*SD*	1.191	1.240	1.395	1.298	1.164	1.511	1.004	1.434	1.276
Digit Span task	Backward	*N*	39	38	39	36	32	36	35	33	36
		*M*	5.667	5.737	6.026	5.972	6.250	6.056	5.971	6.273	6.472
		*SD*	1.420	1.309	1.460	1.341	1.685	1.672	1.740	1.329	1.612
	Forward	*N*	39	38	39	36	32	36	35	34	36
		*M*	6.231	6.211	6.359	6.333	6.281	6.444	6.400	6.265	6.667
		*SD*	1.063	1.018	1.203	1.171	1.397	1.107	1.241	1.463	1.146
Simon task	Congruent	*N*	40	39	39	36	32	36	35	34	36
		*M*	811.787	809.942	680.210	726.375	750.295	644.638	698.373	716.473	602.003
		*SD*	232.453	304.307	166.732	139.661	162.228	149.820	179.273	177.056	114.224
	Incongruent	*N*	40	39	39	36	32	36	35	34	36
		*M*	853.160	867.578	737.168	788.631	808.985	714.171	754.941	770.747	663.365
		*SD*	230.922	247.570	173.335	195.972	175.134	182.647	174.470	189.063	133.611
	Simon effect	*N*	40	39	39	36	32	36	35	34	36
		*M*	69.138	69.603	87.744	47.583	88.563	69.819	44.843	70.382	72.375
		*SD*	87.483	239.714	93.229	84.425	133.390	100.031	92.923	135.303	140.725

*N*—sample size, *M*—mean, *SD*—standard deviation. The mean was calculated from the median of individual scores for each of the conditions.

## Data Availability

The data presented in this study are available on request from the corresponding authors.

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
