# Peer review of "Heliox at 4 ATA Reduces Error Rates Compared to Trimix and Air, but It Does Not Affect Short-Term Memory in Hyperbaric Conditions"

_biology, 2025, doi:10.3390/biology14121748_

Round 1

Reviewer 1 Report

Comments and Suggestions for Authors

The authors address an interesting topic that has already been studied, but that still deserves further data and investigations. Their approach is of interest and enlightens diving physiology. The Simon effect has to my knowledge never been discussed in such situations and adds clear insight.

However, some corrections and suggestions may improve the actual manuscript.

To be corrected

Abstract           -> reflecting practice the effects of practice. The most robust finding….

-> the participants breathing heliox committed had….

Avoid abbreviations in the titles, for instance you cannot just write RT for reaction time, you can on the other hand write: Reaction Time (RT)

I must say that I liked the way that the manuscript is constructed. Every section is well described and clear.

My major concern is about the discussion and the selected choice of the references.

There are some pivotal works in the last years that are not mentioned and just a quick search in pubmed.  I think that some of them are of interest for instance for a more specific differentiation between reaction time and more elaborate tests. Some research articles are comparing Heliox with air and trimix, and other are more dedicated to hyperbaric chamber or “wet” dives.

I believe that the discussion will benefit from some more sentences regarding those findings.

Thank you for letting me review such an interesting manuscript.

Author Response

[Comment 1] Abstract           -> reflecting practice the effects of practice. The most robust finding….-> the participants breathing heliox committed had….

Respond: The abstract has been modified to meet words limit criteria. Mentioned sentenced has been changed.

[Comment 2] Avoid abbreviations in the titles, for instance you cannot just write RT for reaction time, you can on the other hand write: Reaction Time (RT).

Respond: Thank you. Referring to abbreviations in the titles, it has been changed as you suggested.

[Comment 3] My major concern is about the discussion and the selected choice of the references. There are some pivotal works in the last years that are not mentioned and just a quick search in pubmed.  I think that some of them are of interest for instance for a more specific differentiation between reaction time and more elaborate tests. Some research articles are comparing Heliox with air and trimix, and other are more dedicated to hyperbaric chamber or “wet” dives. I believe that the discussion will benefit from some more sentences regarding those findings.

Respond: In relation to your suggestions regarding broadening the reference list including air, heliox and trimix in diving we did again searched available researches and we would like to mark that most of them concern deeper dives, mostly over 100 msw. As breathing mixtures like trimix or heliox are hardly ever used in recreational diving, these articles were not primary listed in the references regarding our research. We agree that expanding the discussion with more references may improve this manuscript. Thank you for this suggestion. We changed the discussion by adding more paragraphs and references.

Reviewer 2 Report

Comments and Suggestions for Authors

This is a manuscript describing a study on how ambient pressure and breathing gas content might influence cognitive performance when assessed using neuropsychological testing.

The study concerns an important topic in diving medicine, where more knowledge is needed.

The manuscript is well-structured and generally easy to follow. It was interesting to read.

But weaknesses and possible confounders associated with the chosen methodology need to be discussed in more detail. A major concern is that the results of the study may be influenced by carryover effects. This is an important issue that must be considered when the results are analysed and interpreted, as it concerns the internal validity of the study and the reliability of the results.

Below are my comments and suggestions, general and more specific:

As I understand the manuscript preparation guidelines, there should be a Simple Summary of no more than 200 words in one paragraph. This item is missing and should be added to the manuscript.

As I understand the instructions for Authors the Abstract should contain up to 200 words and the following headings: Background, Methods, Results, Conclusions. In the present Abstract, there is also a heading named Objectives after Background. The total word count exceeds 200. The authors could consider to shorten the text and remove the heading Objectives.

Row 27: The word practice should be removed.

Row 35: The conclusion “breathing air promoted speed at the cost of mistakes” is not supported by data presented in the Abstracts Results section, which makes the statement seem odd.

The sentence on rows 74-76 (Collectively, ... .... gender.)and the sentence on rows 77-78 (Multiple... ... hyperbaric conditions) are overlapping, i.e. what is said in the latter is already stated in the former. Therefore, the sentence on row 77-78 could be omitted.

Confounding factors associated with psychometric testing are listed on rows 94-97. On this spot in the text, the authors could also consider describing carryover effects associated with psychometric testing.

The statement on rows 99-101 “Hyperbaric oxygen therapy has shown therapeutic promise in reducing neuroinflammation and improving cognition in conditions such as dementia and traumatic brain injury” needs to be supported by good references. The authors provide two references, numbered as 18 and 19.

Reference 18, “Hyperbaric oxygen ameliorates cognitive impairment in patients with Alzheimer’s disease and amnestic mild cognitive impairment”, does not concern patients with traumatic brain injuries but those with Alzheimer´s disease and “amnestic mild cognitive impairment”.

Reference 19, “Anxiety impact on scuba performance and underwater cognitive processing ability” is of no relevance to the authors statement on the effects of hyperbaric oxygen. As indicated by its title, it concerns how anxiety influences performance and cognitive processing, not hyperbaric oxygen.

Therefore, reference 19 should be omitted and at least one, preferably two, additional references, should be presented alongside reference 18.

On rows 114-122 the rationale for the study and underlying hypotheses are presented in a straightforward way.

The authors could consider to move the second half of the first sentence in the paragraph, “…were exposed to dry hyperbaric conditions at a pressure of 4 ATA (equivalent to 30 msw)” (rows 129-130) to the paragraph titled Procedure.

It would be good to know how the study subjects were recruited. Otherwise the risk of selection bias could not be assessed.

When I read the text on rows 137-143, I come to the conclusion that 33 subjects were eligible for analysis. If I have understood it correctly, there were 71 original subjects but 25 did not meet the inclusion criterias or refused participation. A further 7 subjects withdrew during the study, and “Data from six participants were excluded from the analyses because they completed only a single test, which could not be evaluated in the context of the study hypotheses.” If I am correct, this leaves 33 subjects. But later in the manuscript, on rows 168-170, it is written; “In total, 40 participants completed the experiment with air, 36 with heliox (17 women), and 37 with trimix (18 women). Consequently, not all results were available for statistical analyses.” In Table 1, where the number of subjects included in analyses are detailed for each intervention, their number varies between 32 and 40.

I understand that all subjects did not participate in all interventions, but for me the numbers still do not add up. I beg the authors to forgive me if I have misunderstood the text on rows 137-143, but it would be good if they could clarify, preferably with a flow chart, how subjects were lost at inclusion and during the subsequent study process.

On rows 142-143 it is stated that the mean age of the study subjects was 32.54 years with a range from 19-45 years. It seems unnecessary detailed to state the mean age with two decimals. One decimal would suffice. The proper measure of dispersion for a mean value could be standard deviation or variance, but not range. The range is used together with the median value. It could be considered to present both mean and median age, with standard deviation and range, respectively.

The sentence on rows 144-145 could be omitted as it repeats information already stated earlier in the paragraph.

On rows 161-162, it is stated that the different breathing gases where applied “in a randomised order.” The randomisation process used should preferably be described in detail, because the order in which subjects were exposed to different breathing gases may have influenced the results.

On rows 164-165 it says that “information about the gas used was provided to participants upon request after completion of testing.” It would be good if the authors clarified if this information could be provided after each single test or only after the completion of all hyperbaric exposures and all tests.

Section 2.4, Research assumptions, could be omitted as the information in it is already given at the end of the introduction.

The authors state that they used Friedman ANOVA and Wilcoxon signed-rank test for statistical analyses. To my knowledge, both these tests are designed for dependent samples, which is appropriate for analyses comparing data derived from one group before, during and after an intervention. That is what the authors describe in the Results section 3.1 when changes within one group is analysed.

But in Results section 3.2, the authors report data on comparisons between groups exposed to different breathing gases. I may be wrong, but I believe that in such cases subject groups should be considered as independent when compared. And if so, Friedman ANOVA and Wilcoxon signed-rank test are not appropriate to use for inference. Corresponding non-parametric tests for independent variables are for example Kruskal-Wallis and Mann Whitney U, respectively.

I recommend that the authors clearly explain their choices of statistical tests. Advise of a professional statistician could preferably be sought. If a professional statistician already has been consulted, it would be good to state so in the manuscript.

Another important methodological issue that must be addressed when the effects of different breathing gases are analysed is the carryover effect and its potential impact on the results. Other confounders should also be considered. In section 4.1 (rows 494-497) the authors comment briefly on “practice” and “state” effects but they also state that such effects “are unlikely to account for the mixture-specific differences reported here”. I think that carryover effects should be considered when assessing the results obtained in the study.

Each subject went through psychometric testing three times (before, during and after intervention) on three different occasions (breathing air, trimix or heliox). And, on at least the first occasion, a practice trial was administered. Effects of repeated testing (i. e. improved results through practice, increased familiarity with the testing situation and the test themselves, development of cognitive strategies over time, et cetera) might have influenced results and confounded the results.

There were a 24 hour “wash-out” period for breathing gases but how possible effects of repeated psychometric testing (i.e. the training effect) was handled is not mentioned.

It is possible that subjects performed better when tested on the third occasion than they did initially. Therefore, the results of the study might have been different if the effects of breathing gases had been analyzed based on their order of exposure. This is an important issue that must be considered and commented upon by the authors. Otherwise, the validity of the study results and the authors´ conclusions could be questioned.

Potential carryover effects could be assessed and managed in different ways when data are analysed. Again, advise of a professional statistician should preferably be sought. 

The authors comment on “practice” and “state” effects on rows 432-436. But there are other potential confounders that ideally should be commented upon, for example tiredness/fatigue and motivation.

The authors report a considerable loss of study subjects in Section 2.1. About half of the subjects originally intended for inclusion were either not included or lost later on during the study. Of 46 included subjects, 13 (28%) were not fully analysed. Ideally, the authors should comment on the possible effect this might have had on reported data.

The authors present statistically significant differences in psychometric test results after exposure to different breathing gases. It would be good if they also commented on the effect size, i.e. what they consider constitute a minimally relevant difference.

Author Response

[Comment 1] But weaknesses and possible confounders associated with the chosen methodology need to be discussed in more detail. A major concern is that the results of the study may be influenced by carryover effects. This is an important issue that must be considered when the results are analysed and interpreted, as it concerns the internal validity of the study and the reliability of the results.

Respond 1: Thank you. We added explanations in section Limitations.

[Comment 2] As I understand the manuscript preparation guidelines, there should be a Simple Summaryof no more than 200 words in one paragraph. This item is missing and should be added to the manuscript. As I understand the instructions for Authors the Abstract should contain up to 200 words and the following headings: Background, Methods, Results, Conclusions. In the present Abstract, there is also a heading named Objectives after Background. The total word count exceeds 200. The authors could consider to shorten the text and remove the heading Objectives.

Respond 2: The abstract has been modified to meet words limit criteria and the simple summary has been added according to manuscript guidelines.

[Comment 3] Row 27: The word practiceshould be removed.

Respond 3: This word has been removed as the abstract has been changed.

[Comment 4] Row 35: The conclusion “breathing airpromoted speed at the cost of mistakes” is not supported by data presented in the Abstracts Results section, which makes the statement seem odd.

Respond 4: This sentence has been removed from the abstract as well as its sections.

[Comment 5] The sentence on rows 74-76 (Collectively, ... .... gender.)and the sentence on rows 77-78 (Multiple... ... hyperbaric conditions) are overlapping, i.e. what is said in the latter is already stated in the former. Therefore, the sentence on row 77-78 could be omitted.

Respond 5: This sentence has been removed as suggested.

[Comment 6] Confounding factors associated with psychometric testing are listed on rows 94-97. On this spot in the text, the authors could also consider describing carryover effects associated with psychometric testing.

Respond 6: Thank you. We added paragraphs in section Discussion.

[Comment 7] The statement on rows 99-101 “Hyperbaric oxygen therapy has shown therapeutic promise in reducing neuroinflammation and improving cognition in conditions such as dementia and traumatic brain injury” needs to be supported by good references. The authors provide two references, numbered as 18 and 19.

Reference 18, “Hyperbaric oxygen ameliorates cognitive impairment in patients with Alzheimer’s disease and amnestic mild cognitive impairment”, does not concern patients with traumatic brain injuries but those with Alzheimer´s disease and “amnestic mild cognitive impairment”.

Reference 19, “Anxiety impact on scuba performance and underwater cognitive processing ability” is of no relevance to the authors statement on the effects of hyperbaric oxygen. As indicated by its title, it concerns how anxiety influences performance and cognitive processing, not hyperbaric oxygen.

Therefore, reference 19 should be omitted and at least one, preferably two, additional references, should be presented alongside reference 18.

 Respond 7: Thank you. These references has been changed as suggested.

[Comment 8] The authors could consider to move the second half of the first sentence in the paragraph, “…were exposed to dry hyperbaric conditions at a pressure of 4 ATA (equivalent to 30 msw)” (rows 129-130) to the paragraph titled Procedure.

Respond 8: The half of this sentence has been moved to section Procedure following your suggestion.

[Comment 9]  It would be good to know how the study subjects were recruited. Otherwise the risk of selection bias could not be assessed. 

Respond 9: Thank you. This issue has been described in subsection Participants.

[Comment 10] When I read the text on rows 137-143, I come to the conclusion that 33 subjects were eligible for analysis. If I have understood it correctly, there were 71 original subjects but 25 did not meet the inclusion criterias or refused participation. A further 7 subjects withdrew during the study, and “Data from six participants were excluded from the analyses because they completed only a single test, which could not be evaluated in the context of the study hypotheses.” If I am correct, this leaves 33 subjects. But later in the manuscript, on rows 168-170, it is written; “In total, 40 participants completed the experiment with air, 36 with heliox (17 women), and 37 with trimix (18 women). Consequently, not all results were available for statistical analyses.” In Table 1, where the number of subjects included in analyses are detailed for each intervention, their number varies between 32 and 40.

 I understand that all subjects did not participate in all interventions, but for me the numbers still do not add up. I beg the authors to forgive me if I have misunderstood the text on rows 137-143, but it would be good if they could clarify, preferably with a flow chart, how subjects were lost at inclusion and during the subsequent study process.

Response 10: Thank you for this suggestion. As we understand this text could cause confusion, so we have changed a description of our study group. Please see subsection Participant.

We gained 46 subjects  who have been tested, but we collected complete data from 27 individuals. Seven subjects resigned during our research without giving a reason that is in line with bioethical principles. Some subject did not complete entire test (3 subjects did not complete test on air, 4 subjects did not complete it on heliox, 5 subjects did not complete it on trimix). Therefore, we do not have results of all examined participants.  This data was included, but during individual statistical tests, all missing data were eliminated on the fly.

[Comment 11] On rows 142-143 it is stated that the mean age of the study subjects was 32.54 years with a range from 19-45 years. It seems unnecessary detailed to state the mean age with two decimals. One decimal would suffice. The proper measure of dispersion for a mean value could be standard deviation or variance, but not range. The range is used together with the median value. It could be considered to present both mean and median age, with standard deviation and range, respectively.

Respond 11: Thank you. This description has been corrected.

[Comment 12] The sentence on rows 144-145 could be omitted as it repeats information already stated earlier in the paragraph.

Respond 12: Repeated information has been removed.

[Comment 13] On rows 161-162, it is stated that the different breathing gases where applied “in a randomised order.” The randomisation process used should preferably be described in detail, because the order in which subjects were exposed to different breathing gases may have influenced the results.

Respond 13: The group of qualified volunteers was presented with the first two dates, which had already been determined by the researchers. Each participant could choose preferable day of tests. The last test with the missing breathing gas was selected according to participant’s availability. The order of breathing mixtures used in the hyperbaric chamber was randomized in first two stages. For safety reason, the researcher and the attendant had to know the breathing gas used in the chamber to follow precautions. The double-blinding of the trial was not possible for this reason.

[Comment 14] On rows 164-165 it says that “information about the gas used was provided to participants upon request after completion of testing.” It would be good if the authors clarified if this information could be provided after each single test or only after the completion of all hyperbaric exposures and all tests.

Respond 14: The information about breathing mixture used in each test was provided after completing all three stages upon participant’s request.

[Commet 15] Section 2.4, Research assumptions, could be omitted as the information in it is already given at the end of the introduction.

Respond 15: This section has been removed as suggested.

[Comment 16] The authors state that they used Friedman ANOVA and Wilcoxon signed-rank test for statistical analyses. To my knowledge, both these tests are designed for dependent samples, which is appropriate for analyses comparing data derived from one group before, during and after an intervention. That is what the authors describe in the Results section 3.1 when changes within one group is analysed.

But in Results section 3.2, the authors report data on comparisons between groups exposed to different breathing gases. I may be wrong, but I believe that in such cases subject groups should be considered as independent when compared. And if so, Friedman ANOVA and Wilcoxon signed-rank test are not appropriate to use for inference. Corresponding non-parametric tests for independent variables are for example Kruskal-Wallis and Mann Whitney U, respectively.

I recommend that the authors clearly explain their choices of statistical tests. Advise of a professional statistician could preferably be sought. If a professional statistician already has been consulted, it would be good to state so in the manuscript.

Respond 16: Our study was conducted in one group. The same group of volunteers performed tests at three different stages. Obtained data was analyzed among this one group, even with three different breathing mixtures. Therefore, these results meet the criteria for dependent samples. In opinion of our statistician, chosen tests (Friedman and Wilcoxon signed-rank tests) are appropriate for this analysis.

[Comment 17] Another important methodological issue that must be addressed when the effects of different breathing gases are analysed is the carryover effect and its potential impact on the results. Other confounders should also be considered. In section 4.1 (rows 494-497) the authors comment briefly on “practice” and “state” effects but they also state that such effects “are unlikely to account for the mixture-specific differences reported here”. I think that carryover effects should be considered when assessing the results obtained in the study.

 Each subject went through psychometric testing three times (before, during and after intervention) on three different occasions (breathing air, trimix or heliox). And, on at least the first occasion, a practice trial was administered. Effects of repeated testing (i. e. improved results through practice, increased familiarity with the testing situation and the test themselves, development of cognitive strategies over time, et cetera) might have influenced results and confounded the results.

There were a 24 hour “wash-out” period for breathing gases but how possible effects of repeated psychometric testing (i.e. the training effect) was handled is not mentioned.

It is possible that subjects performed better when tested on the third occasion than they did initially. Therefore, the results of the study might have been different if the effects of breathing gases had been analyzed based on their order of exposure. This is an important issue that must be considered and commented upon by the authors. Otherwise, the validity of the study results and the authors´ conclusions could be questioned.

 Potential carryover effects could be assessed and managed in different ways when data are analysed. Again, advise of a professional statistician should preferably be sought. 

Respond 17: Thank you for your suggestion. Learning effect has been described also in improved section Discussion section. In reference to the article, Simon effect reduces with learning effect, but it returns after short breaks. Therefore, in our study with mandatory breaks over 24 hours, the influence of study effect remains slight.

[Comment 18] The authors comment on “practice”and “state” effects on rows 432-436. But there are other potential confounders that ideally should be commented upon, for example tiredness/fatigue and motivation.

Respond 18: We have described this issue in in section Limitations.

[Comment 19] The authors report a considerable loss of study subjects in Section 2.1. About half of the subjects originally intended for inclusion were either not included or lost later on during the study. Of 46 included subjects, 13 (28%) were not fully analysed. Ideally, the authors should comment on the possible effect this might have had on reported data.

Respond 19: We thank the reviewer for this observation. Several participants withdrew from the study before completing all experimental stages. The main reasons for withdrawal included scheduling conflicts, the administrative burden of participation, and the demanding nature of multiple hyperbaric testing sessions. These factors were organizational rather than performance-related and therefore unlikely to have systematically affected cognitive outcomes. Although the resulting reduction in sample size may have slightly decreased the statistical power, it is not expected to have biased the reported results. This limitation has been acknowledged in the revised Participants subsection

[Comment 20] The authors present statistically significant differences in psychometric test results after exposure to different breathing gases. It would be good if they also commented on the effect size, i.e. what they consider constitute a minimally relevant difference.

Respond 20: Thank you for this suggestion. We have considered the effect sizes for all reported comparisons. Overall, the Friedman tests indicated small effects, whereas post-hoc Wilcoxon contrasts showed medium effects, particularly in comparisons involving the air condition. Effect sizes were calculated using Kendall’s W for Friedman tests and r = Z/√N for Wilcoxon pairwise comparisons. As a minimally relevant difference, we used a pragmatic within-subject threshold roughly corresponding to a medium effect (≈0.3 in r or ≈0.5 SD), which was exceeded mainly in contrasts involving air vs. heliox/trimix. Detailed effect size values are not included in the manuscript, but the observed differences were assessed against this threshold.

Round 2

Reviewer 2 Report

Comments and Suggestions for Authors

Dear authors!

Thank you for the revised version of your manuscript "Heliox at 4 ATA reduces error rates compared to trimix and air, but it does not affect short-term memory in hyperbaric conditions"

I have only a few further considerations, which are as follows: 

On rows 22-23 you write "Assuming the appearance of inert gas narcosis symptoms at depths equivalent to 2-2.5 ATA (atmospheres absolute)..."

The sentence is technically correct, but I believe that many people would say that signs of inert gas (nitrogen) narcosis usually become evident at higher ambient pressures than 2-2.5 ATA, although some few individuals may react to increased nitrogen pressures as as early as at 2 ATA. The pressures 3.5-4 ATA are sometimes used to describe when, in general, symptoms of inert gas narcosis become observable. You could consider rephrasing the sentence. 

It should be written "...2-2.5 atmosphere absolute (ATA)..." , as this is the first time you use the abbreviation ATA. Therefore ATA should be in brackets. But, maybe better, ATA could be omitted.

The sentence "During the study, seven participants withdrew without providing a reason, and 25 either did not meet the requirements or declined to accept the study conditions" on rows 161-162 should preferably be  omitted as this information is already given on rows 155-157.

It is mentioned in the title of the manuscript that breathing Heliox does not affect short-term memory in hyperbaric conditions. This result is described in section 3.1.2 "Attention and Memory Performance". As you choose to mention short-term memory in your title, it would be good if you also mentioned the preserved short-term memory when breathing Heliox in your Conclusion, i.e. the final paragraph.
